# EquiGraspFlow: SE(3)-Equivariant 6-DoF Grasp Pose Generative Flows

**Byeongdo Lim**[*1]    **Jongmin Kim**[*1]    **Jihwan Kim**[1]    **Yonghyeon Lee**[2]    **Frank C. Park**[1,3]

[1]Seoul National University    [2]Korea Institute for Advanced Study (KIAS)    [3]SAIGE

{bdlim, jmkim, jihwankim}@robotics.snu.ac.kr    ylee@kias.re.kr    fcp@snu.ac.kr

**Abstract:** Traditional methods for synthesizing 6-DoF grasp poses from 3D observations often rely on geometric heuristics, resulting in poor generalizability, limited grasp options, and higher failure rates. Recently, data-driven methods have been proposed that use generative models to learn the distribution of grasp poses and generate diverse candidate poses. The main drawback of these methods is that they fail to achieve SE(3)-equivariance, meaning that the generated grasp poses do not transform correctly with object rotations and translations. In this paper, we propose *EquiGraspFlow*, a flow-based SE(3)-equivariant 6-DoF grasp pose generative model that can learn complex conditional distributions on the SE(3) manifold while guaranteeing SE(3)-equivariance. Our model achieves the equivariance without relying on data augmentation, by using network architectures that guarantee it by construction. Extensive experiments show that *EquiGraspFlow* accurately learns grasp pose distribution, achieves the SE(3)-equivariance, and significantly outperforms existing grasp pose generative models. Code is available at https://github.com/bdlim99/EquiGraspFlow.

**Keywords:** 6-DoF grasp pose generation, equivariance, generative models, continuous normalizing flows

## 1 Introduction

Synthesizing six degrees of freedom (DoF) grasp poses from 3D observations of an object (e.g., surface point cloud) is a fundamental task in robotics. Approaches that produce a limited number of grasp poses [1, 2, 3] carry a high risk of grasping failures, especially in constrained environments. For example, obstacles or kinematic constraints may prevent the robot from reaching certain candidate grasp poses; therefore, the more diverse candidates, the higher the probability of grasp success.

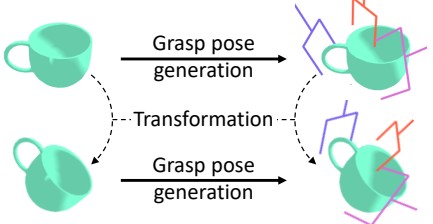

Figure 1: An example of SE(3)-equivariant grasp pose generation: the generated grasp poses identically transform with the object.

Earlier approaches to generate diverse grasp poses rely on geometric heuristics to suggest multiple candidates based on an object's geometry [4, 5]. These methods typically identify antipodal surface points – pairs of surface points with opposite surface normals – as the contact points of the gripper's fingertips. However, their reliance on antipodal points limits the diversity of the grasp poses and makes the methods sensitive to observational noise in surface points and normals.

Recently, data-driven approaches have gained significant attention. These methods first generate sufficiently diverse grasp poses through simulation [6], then use this data to learn the distribution of grasp poses and train grasp pose generative models [7, 8]. The trained models not only generate diverse poses but can also be made robust to observational noise by employing domain randomization.

---

[*]Equal contribution

8th Conference on Robot Learning (CoRL 2024), Munich, Germany.

Mousavian et al. [7] employed the Variational Autoencoder (VAE) [9] as a grasp pose generative model, while Urain et al. [8] more recently trained an energy-based model via the denoising score matching [10], demonstrating improved performance in learning complex grasp pose distribution.

However, the primary flaw with existing grasp pose generative models is that they do not produce consistent grasp poses for rotated objects, leading to significant failure in some cases. An ideal model should generate grasp poses that transform identically for rotated and translated objects, as shown in Figure 1. Such models are considered SE(3)-equivariant. Existing methods fail to ensure this equivariance, despite efforts to achieve it through data augmentation. Equivariant models, which have drawn significant attention from robot learning [11, 12], provide a promising solution to this issue, enabling robust and accurate grasp pose generation.

Our main contribution is *EquiGraspFlow*, a SE(3)-equivariant 6-DoF grasp pose generative model where the equivariance is guaranteed by the network architectures, hence no data augmentation is required. Specifically, we adopt the Continuous Normalizing Flows (CNFs) framework [13, 14, 15], which, by utilizing time-dependent velocity fields as an infinite number of infinitesimally small transformations, learns complex distributions more effectively than diffusion models or discrete normalizing flows. We then formulate the necessary conditions to guarantee the SE(3)-equivariance of grasp pose generation for CNF on SE(3) conditioned on point cloud input.

Among these conditions, a non-trivial one requiring careful consideration is SE(3)-equivariance of the time-dependent velocity fields. Our method leverages the Vector Neurons (VNs) [16], which is a popular choice in constructing SE(3)-equivariant point cloud networks. Although VNs have shown remarkable success, it allows only vector variables as inputs and cannot accommodate scalar variables such as time. This limitation makes it challenging to use VNs for constructing SE(3)-equivariant time-dependent velocity fields. To address this, we introduce a novel *equivariant lifting layer* that equips scalar variables with an equivariant vector basis, enabling the effective use of VNs in constructing SE(3)-equivariant time-dependent velocity fields.

Through experiments conducted in both simulation and real-world environments, we have validated that our method surpasses existing 6-DoF grasp pose generative models [7, 8]. Notably, our method achieves superior results and consistent performance, even with changes in the object's rotation, without employing any augmentation strategies for equivariance. Real-world experiments validate that our method seamlessly applies to real-world applications.

## 2 Related Work

Synthesizing 6-DoF grasp poses is an active research area tackled with numerous methods [17, 18, 19]. Specifically, we focus on generative model-based approaches [7, 8, 20] and methods that guarantee equivariance [21, 22, 23], which are of particular relevance to our work. Additionally, we present generative models that incorporate equivariance across various domains.

**Generative models for grasping** Mousavian et al. [7] develop 6-DOF GraspNet, a grasp pose generative model based on VAE [9], refining the generated grasp poses using an additional data-driven grasp evaluator. Urain et al. [8] propose a diffusion model [24] on the SE(3) manifold, termed SE(3)-DiffusionFields, for grasp pose generation. Weng et al. [20] develop CAPGrasp, an $\mathbb{R}^3 \times SO(2)$-equivariant grasp pose generative model under the assumption of approach-constrained grasp. Despite these advancements, existing grasp pose generative models have not fully explored SE(3)-equivariance; they rely on augmentation strategies or achieve only partial equivariance under the assumption of the grasp. In contrast, our method fully ensures the SE(3)-equivariance without relying on any assumptions or augmentation strategies.

**Equivariance for grasping** Zhu et al. [21, 22] incorporate SE(2)-equivariance for generating planar grasp poses from top-down image observations, achieving enhanced sample efficiency. Huang et al. [23] construct Edge Grasp Network, an SE(3)-invariant grasp quality function, resulting in improved grasp quality prediction performance for 6-DoF grasp poses. However, because these

methods do not utilize generative models, they either produce only a limited set of grasp poses or rely on geometric heuristics that lack sufficient diversity. Unlike these methods, our method employs a generative model to produce a diverse set of grasp poses while ensuring full SE(3)-equivariance.

**Equivariant Generative Models**  Katsman et al. [25] introduce equivariant generative models on manifolds by applying equivariant manifold flows. Chen et al. [26] propose an SO(2)-equivariant conditional generative model for generating 2D trajectories in Euclidean space, and Rozenberg and Freedman [27] propose an E(3)- and permutation-equivariant conditional generative model for generating 3D graphs in Euclidean space. Additionally, Zwartsenberg et al. [28] develop a permutation-equivariant conditional generative model for generating traffic scenes and object bounding boxes in Euclidean space. These methods primarily focus on learning distributions either unconditional or in Euclidean space, making them difficult to apply to SE(3)-equivariant 6-DoF grasp pose generation, which requires conditional distributions on the SE(3) manifold. To address this limitation, we propose an SE(3)-equivariant conditional generative model on the SE(3) manifold.

## 3   Preliminaries: Continuous Normalizing Flows on SE(3)

As preliminaries, we introduce the Continuous Normalizing Flows (CNFs) [13, 14, 15] tailored for SE(3) cases. We begin by defining key notations; we denote an element of SE(3) by $T = (R, x)$, where $R \in \mathrm{SO}(3)$ and $x \in \mathbb{R}^3$, and refer to it as a pose or grasp pose. Given a three-dimensional vector $a = (a_1, a_2, a_3)$, $[a]$ is a $3 \times 3$ skew symmetric matrix with entries defined as $[a]_{12} = -a_3, [a]_{13} = a_2, [a]_{23} = -a_1$. By decomposing flows on SE(3) into SO(3) and $\mathbb{R}^3$, we construct time-dependent angular and linear velocity fields, $\omega : [0, 1] \times \mathrm{SE}(3) \to \mathbb{R}^3$ and $v : [0, 1] \times \mathrm{SE}(3) \to \mathbb{R}^3$, which map a time and a pose to angular and linear velocity vectors, respectively. We collectively define these two velocity fields as a time-dependent vector field $u : [0, 1] \times \mathrm{SE}(3) \to \mathbb{R}^3$.

**Unconditional Continuous Normalizing Flows on SE(3)**  The CNF on SE(3) models a target distribution $q(T)$ by transforming a prior distribution $p_0(T)$ using the time-dependent angular and linear velocity fields $\omega_\theta$ and $v_\phi$ where $\theta$ and $\phi$ are trainable parameters. This transformation is guided by the following ordinary differential equations (ODEs)

$$\dot{R} = [\omega_\theta(t, T)]R, \quad \dot{x} = v_\phi(t, T). \tag{1}$$

It generates a flow and a probability density path $p_t(T)$ for time $t \in [0, 1]$. Sampling from $p_\tau(T)$ involves sampling from the prior $p_0(T)$ and transforming these initial samples along the flow by solving the ODEs over $t \in [0, \tau]$. The parameters $\theta$ and $\phi$ are trained to ensure that the transformed distribution $p_1(T)$ matches the target distribution $q(T)$.

**Conditional Continuous Normalizing Flows on SE(3)**  To model a target conditional distribution $q(T|c)$ for some condition variable $c$, we use a prior conditional distribution $p_0(T|c)$ and velocity fields conditioned on $c$; the ODEs become

$$\dot{R} = [\omega_\theta(t, c, T)]R, \quad \dot{x} = v_\phi(t, c, T) \tag{2}$$

which lead to a conditional probability density path $p_t(T|c)$ for $t \in [0, 1]$. In our setting, the condition variable $c$ represents a point cloud of an object and the target conditional distribution $q(T|c)$ corresponds to the distribution of successful grasp poses for that object.

## 4   EquiGraspFlow: SE(3)-Equivariant 6-DoF Grasp Pose Generative Flows

In this section, we introduce *EquiGraspFlow*, which generates diverse 6-DoF grasp poses from point cloud inputs using the conditional CNF approach while ensuring SE(3)-equivariance. Denoting a point cloud by $\mathcal{P} = \{x_k \in \mathbb{R}^3\}_{k=1}^K$, EquiGraspFlow utilizes the time-dependent conditional velocity fields $\omega_\theta(t, \mathcal{P}, T), v_\phi(t, \mathcal{P}, T)$ and a prior conditional distribution $p_0(T|\mathcal{P}) = p_0(R)p_0(x|\mathcal{P})$, where

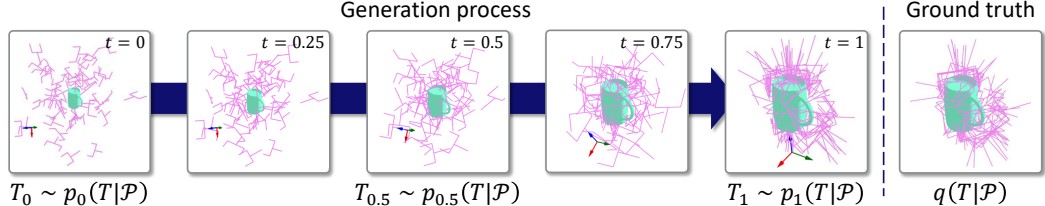

Figure 2: Grasp pose generation process of EquiGraspFlow.

$p_0(R)$ is uniform over $\mathrm{SO}(3)$ and $p_0(x|\mathcal{P})$ is Gaussian in $\mathbb{R}^3$ with its mean located at the center of the point cloud. As illustrated in Figure 2, the model transforms $p_0(T|\mathcal{P})$ into $p_1(T|\mathcal{P})$ through a flow constructed from the velocity fields, as detailed in Figure 3.

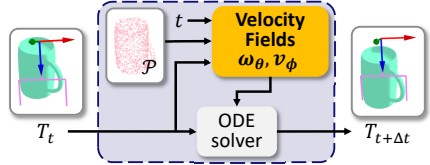

Figure 3: Construction of the flow from $t$ to $t + \Delta t$ using the velocity fields $\omega_\theta$, $v_\phi$ and an ODE solver.

We use a dataset $\mathcal{D} := \{(\mathcal{P}_i, \{T_{ij}\}_{j=1}^{M_i})\}_{i=1}^N$, consisting of pairs of the point cloud $\mathcal{P}_i$ of the $i$-th object and its corresponding set of successful grasp poses $\{T_{ij}\}_{j=1}^{M_i}$. We assume that each set of grasp poses is sampled from the respective ground-truth conditional distribution $q(T|\mathcal{P}_i)$. The neural velocity fields $\omega_\theta$ and $v_\phi$ are trained with $\mathcal{D}$ to ensure that the transformed conditional distribution $p_1(T|\mathcal{P}_i)$ closely approximate $q(T|\mathcal{P}_i)$. We employ a Flow Matching framework [29, 30] to train the velocity fields and Guided Flows [31] to enhance the sample quality. Details are provided in Appendix B.2.

In the following section, we explain the incorporation of $\mathrm{SE}(3)$-equivariance into EquiGraspFlow. First, we will establish $\mathrm{SE}(3)$-invariance of conditional distributions, which is essential for $\mathrm{SE}(3)$-equivariant grasp pose generation. We will then derive the necessary conditions to ensure that the transformed conditional distribution $p_1(T|\mathcal{P})$ is $\mathrm{SE}(3)$-invariant.

### 4.1 SE(3)-Invariant Conditional Distributions

We first define transformations for point clouds, grasp poses, and three-dimensional vectors. Given an element $T' = (R', x') \in \mathrm{SE}(3)$, a point cloud $\mathcal{P} = \{x_k\}$ is transformed to $T'\mathcal{P} := \{R'x_k + x'\}$, a grasp pose $T = (R, x)$ is transformed to $T'T = (R'R, R'x + x')$, and a three-dimensional vector $a$ is transformed to $R'a$. Next, we define $\mathrm{SE}(3)$-invariance of conditional distributions:

**Definition 1.** *A distribution on* $\mathrm{SE}(3)$ *conditioned on a point cloud, denoted by* $p(T|\mathcal{P})$*, is* $\mathrm{SE}(3)$-*invariant if* $p(T'T|T'\mathcal{P}) = p(T|\mathcal{P})$ *for any* $T' \in \mathrm{SE}(3)$.

This is a formal description, in the language of probability distributions, that leads to $\mathrm{SE}(3)$-equivariant grasp pose generation. For a transformed point cloud, equivalently transformed grasp poses have the same conditional likelihood as before the transformation.

It is well-established mathematical fact that distributions transformed from an invariant prior via equivariant maps remain invariant [25, 26, 27, 28, 32, 33]. However, existing works primarily focus on distributions either in Euclidean space, unconditional, or both, making them unsuitable for grasp pose generation, which requires conditional distributions on the $\mathrm{SE}(3)$ manifold. To address this, we extend these works to model invariant conditional distributions on $\mathrm{SE}(3)$, thereby satisfying the required conditions. We thus define $\mathrm{SE}(3)$-equivariance of time-dependent conditional vector fields on $\mathrm{SE}(3)$:

**Definition 2.** *A time-dependent vector field on* $\mathrm{SE}(3)$ *conditioned on a point cloud, denoted by* $u(t, \mathcal{P}, T)$*, is* $\mathrm{SE}(3)$-*equivariant if* $u(t, T'\mathcal{P}, T'T) = R'u(t, \mathcal{P}, T)$ *for any* $T' = (R', x') \in \mathrm{SE}(3)$.

This geometric condition is essential for modeling invariant conditional distributions via CNFs on $\mathrm{SE}(3)$. Starting from an invariant prior conditional distribution, equivariant velocity fields preserve the invariance of transformed conditional distributions over time by the following proposition.

**Proposition 1.** *Suppose a prior conditional distribution* $p_0(T|\mathcal{P})$ *is* $\mathrm{SE}(3)$-*invariant. If the time-dependent angular and linear velocity fields* $\omega, v$ *are* $\mathrm{SE}(3)$-*equivariant, then the transformed con-*

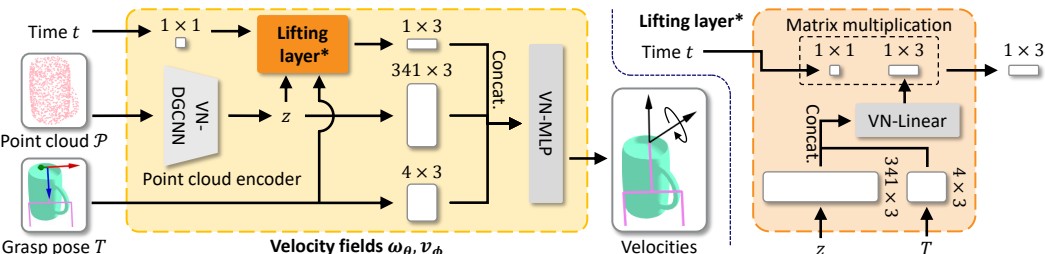

Figure 4: Structure of the velocity fields $\omega_\theta$, $v_\phi$ and the lifting layer. The VN-DGCNN encodes the point cloud $\mathcal{P}$ into a representation $z$ consisting of 341 three-dimensional vectors. The lifting layer uses this representation $z$ along with the grasp pose $T$ to lift the time variable $t$ to a three-dimensional vector. Finally, the VN-MLP takes as input the concatenated list of the lifted time, representation, and grasp pose, and outputs the angular and linear velocities.

*ditional distribution $p_t(T|\mathcal{P})$ at any time $t \geq 0$, defined via the flow of ODEs $\dot{R} = [\omega(t, \mathcal{P}, T)]R$ and $\dot{x} = v(t, \mathcal{P}, T)$, is SE(3)-invariant.*

It is trivial to show that our prior conditional distribution $p_0(T|\mathcal{P})$ is SE(3)-invariant. In the subsequent section, we propose a neural network architecture that guarantees the remaining conditions in Proposition 1: the SE(3)-equivariance of the angular and linear velocity fields.

### 4.2   SE(3)-Equivariant Time-Dependent Conditional Velocity Field Networks

We separate the SE(3)-equivariance into the equivariances on $\mathbb{R}^3$ and SO(3). The $\mathbb{R}^3$-equivariance is achieved by subtracting the point mean $\mu = \sum_k x_k / K$ from $\mathcal{P}$ and $x$ in $T$. The SO(3)-equivariance is achieved by adopting the Vector Neuron (VN) architectures [16], which are designed to be SO(3)-equivariant. The structure of the velocity fields is depicted in Figure 4.

However, directly using the VN architectures is not straightforward since they require lists of three-dimensional vectors as input. Among the inputs of the time-dependent conditional velocity fields, the point cloud $\mathcal{P}$ is a set of three-dimensional vectors, and the pose $T$ can be represented as a list of three-dimensional vectors $(R_1, R_2, R_3, x)$, where $R_i$ is the $i$-th column vector of $R$. However, the time $t$ is a scalar variable, making its incorporation into the VN architectures challenging. To use VNs with minimal modification, we propose an equivariant lifting layer that converts any scalar variables into three-dimensional equivariant vectors, so that they can be concatenated to the list of vectors while maintaining the equivariance of the VN architectures.

Consider a list of $C_1$ scalar variables represented as a column vector $s \in \mathbb{R}^{C_1 \times 1}$ and list of $C_2$ three-dimensional vectors represented as a matrix $V \in \mathbb{R}^{C_2 \times 3}$ – in our case, the time variable corresponds to the case when $C_1 = 1$. We propose a lifting layer, a mapping $f_{\text{lift}} : \mathbb{R}^{C_1 \times 1} \times \mathbb{R}^{C_2 \times 3} \to \mathbb{R}^{C_1 \times 3}$, and consider it to be SO(3)-equivariant if it satisfies $f_{\text{lift}}(s, VR^T) = f_{\text{lift}}(s, V)R^T$ for any $R \in \text{SO}(3)$.

We construct the equivariant lifting layer as $f_{\text{lift}}(s, V) = s f_{\text{equi}}(V)$ where $f_{\text{equi}} : \mathbb{R}^{C_2 \times 3} \to \mathbb{R}^{1 \times 3}$ is any equivariant mapping, i.e., $f_{\text{equi}}(VR^T) = f_{\text{equi}}(V)R^T$. This implies that $f_{\text{equi}}$ produces an equivariant vector from the input vectors $V$, and this equivariant vector is subsequently scaled by each of the $C_1$ scalar variables in $s$, resulting in list of $C_1$ vectors. It is trivial to show that this construction leads to the SO(3)-equivariance of $f_{\text{lift}}$. For $f_{\text{equi}}$, we use the VN architecture [16]. Finally, with the proposed equivariant lifting layer, we can construct an equivariant neural network for the time-dependent conditional velocity fields $\omega_\theta(t, \mathcal{P}, T)$ and $v_\phi(t, \mathcal{P}, T)$.

## 5   Experiments

### 5.1   Experiment Settings

**Baselines**   We compare our model with existing grasp pose generative models, 6-DOF GraspNet [7, 34] and SE(3)-DiffusionFields [8]. To exclusively compare the generation performance of the generative models, we exclude the grasp evaluator used in 6-DOF GraspNet. SE(3)-DiffusionFields

| Augmentation strategy | None | | | | SO(3)-*aug* | | | |
|---|---|---|---|---|---|---|---|---|
| Object category | Laptop | Mug | Bowl | Pencil | Laptop | Mug | Bowl | Pencil |
| 6-DOF GraspNet [7] | 0.7990 | 0.8285 | 1.0072 | 0.5216 | 0.4666 | 0.6683 | 0.6550 | 0.4025 |
| PoiNt-SE(3)-Dif [8] | 0.7184 | 0.7648 | 0.8775 | 0.5530 | 0.6054 | 0.6164 | 0.5047 | 0.5846 |
| EquiGraspFlow (Ours) | **0.3579** | **0.4804** | **0.3114** | **0.2988** | **0.3417** | **0.4828** | **0.3126** | **0.2872** |

(a) Average EMD (lower values are better)

| Augmentation strategy | None | | | | SO(3)-*aug* | | | |
|---|---|---|---|---|---|---|---|---|
| Object category | Laptop | Mug | Bowl | Pencil | Laptop | Mug | Bowl | Pencil |
| 6-DOF GraspNet [7] | 32.96 | 32.58 | 40.93 | 59.35 | 59.11 | 43.64 | 59.67 | 88.71 |
| PoiNt-SE(3)-Dif [8] | 13.03 | 19.14 | 28.87 | 52.33 | 95.65 | 80.98 | 99.87 | 97.85 |
| EquiGraspFlow (Ours) | **99.72** | **90.56** | **100.00** | **98.98** | **99.90** | **92.26** | **100.00** | **99.71** |

(b) Average grasp success rate (%) (higher values are better)

Table 1: EMD and grasp success rate for various objects in simulation experiments.

include two versions: SE(3)-Dif, which uses predefined object's shape codes, and PoiNt-SE(3)-Dif, which encodes object's point cloud into a latent code. To compare with our model which also encodes object's point cloud, we choose PoiNt-SE(3)-Dif as a baseline. Since the baselines can incorporate $\mathbb{R}^3$-equivariance by simply subtracting the point mean from the point cloud and adding the mean back to the generated grasp poses, we evaluate how effectively these models can achieve SO(3)-equivariance through data augmentation.

**Dataset**   We utilize a dataset obtained from the Laptop, Mug, Bowl, and Pencil categories of the ACRONYM dataset [6], comprising 175 laptops, 94 mugs, 38 bowls, and 82 pencils, each with poses configured for grasping by Franka Panda gripper. Our objective is to train a general model capable of grasping every type of object, so we utilize a unified dataset containing all object types to train a single model. The objects are provided in mesh form; therefore, we uniformly sample 1024 points on the mesh surface to obtain a full point cloud. Experiments utilizing partial point clouds are provided in Appendix C.2. For the data augmentation of the training dataset, we construct two strategies: *None* denotes no augmentation, and SO(3)-*aug* denotes augmenting by random arbitrary rotation in SO(3). The validation and test datasets are augmented with evenly sampled rotations using Super-Fibonacci Spirals algorithm [35].

**Evaluation Metrics**   The evaluation metrics we utilize are Earth Mover's Distance (EMD) [36] and grasp success rate. The EMD measures the distance between the distributions of the generated and ground-truth grasp poses, defined by the minimum geodesic distance on the SE(3) manifold required to align the samples. The grasp success rate is assessed by determining whether the Franka Panda gripper successfully holds the object following the grasping action. Both metrics are first averaged across the rotations for each object, and then averaged across all objects.

## 5.2   Simulation Experiments

We conduct two types of experiments in the Nvidia Isaac Gym simulator [37]. The first experiment measures the average EMD and grasp success rate, while the second experiment assesses the consistency of the EMD and grasp success rate with changes in the object's rotation. In the first experiment, the test dataset is augmented with three rotations, while in the second experiment, it is augmented with ten rotations. To measure the grasp success rate, we generate 100 grasp poses in the first experiment and 25 grasp poses in the second experiment. To eliminate inconsistency arising from sampling the prior distribution in the second experiment, we fix initial samples in PoiNt-SE(3)-Dif [8] and EquiGraspFlow, and then rotate them along with the object.

The average EMD and grasp success rate for various objects under different augmentation strategies are presented in Table 1. The existing methods do not account for SO(3)-equivariance, leading

to insufficient performance with high EMD and low grasp success rate in the *None* setting. Conversely, EquiGraspFlow incorporates SO(3)-equivariance, improving performance in learning the grasp pose distribution and generating graspable poses in this setting, as evidenced by lower EMD and higher grasp success rate. Even when arbitrary rotations augment the training dataset in the SO(3)-*aug* setting, there exist discrepancies in the EMD and grasp success rate values between EquiGraspFlow and the baselines, underscoring our method's superior performance. Notably, unlike SE(3)-DiffusionFields, which requires additional data to learn the signed distance function, our model achieves superior performance without additional data or modules. Additionally, by more accurately learning the grasp pose distribution, our model generates more diverse grasp poses than the baselines, as depicted in Figure 6.

Figure 5 and Figure 6 illustrate the consistency of performance as the object rotation varies for models trained with the SO(3)-*aug* setting. Figure 5 presents the average EMD and grasp success rate, with standard deviations indicated by error bars. The standard deviations are calculated with respect to the object rotation and are averaged across all objects. Figure 6 illustrates the generated grasp poses alongside EMD and grasp success rate values for various object rotations, where both the object and the generated grasp poses are inversely rotated to align all scenes. As indicated by the error bars in Figure 5 and the variance of values in Figure 6, our model exhibits more consistent performance compared to the baselines, attributed to its incorporation of SO(3)-equivariance. Notably, our model shows zero standard deviations in Figure 5 and maintains an identical value regardless of the object's rotation in Figure 6, demonstrating the perfect equivariance of our model.

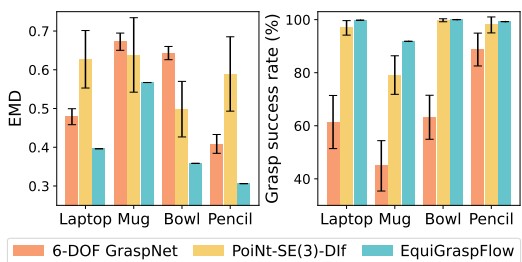

Figure 5: Average and standard deviation of EMD and grasp success rate for models trained with the SO(3)-*aug* setting.

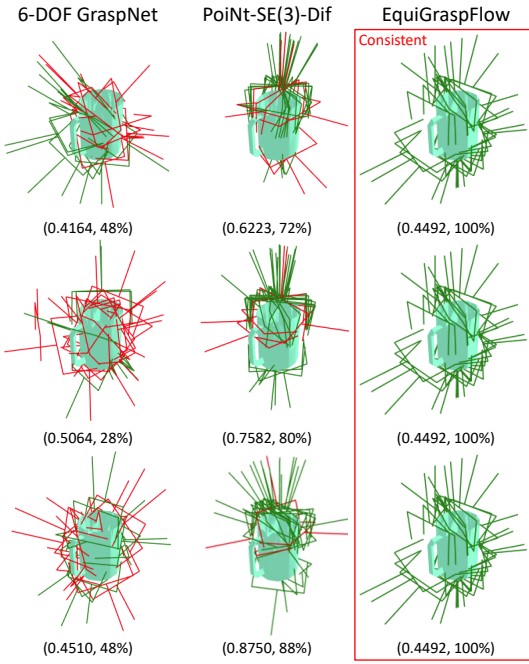

Figure 6: Grasp poses generated by models trained with the SO(3)-*aug* setting for rotated objects. Both the object and grasp poses are inversely rotated to align all scenes. Green indicates successful grasps, while red indicates failures. The EMD and grasp success rate for each scene are annotated below.

## 5.3 Real-World Experiments

In the real-world experiments, we evaluate the grasp success rate of the baselines and our model trained with the SO(3)-*aug* setting. Two mugs and two bowls as shown in Figure 7 are used. For the mugs, the handles are randomly oriented to test grasping, and both mugs are further rotated 90 degrees along the horizontal axis for additional testing, indicated as (R) in Table 2. The bowls are placed on the desk and also rotated at a certain angle along the horizontal axis, also indicated as (R) in Table 2.

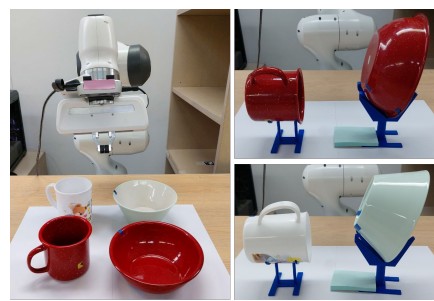

Figure 7: The mugs and bowls used in the real-world experiments along with the Franka Panda robot. The right side of the image shows the mugs and bowls in rotated poses.

| Object | Mug 1 | Mug 1 (R) | Mug 2 | Mug 2 (R) | Bowl 1 | Bowl 1 (R) | Bowl 2 | Bowl 2 (R) | Total |
|---|---|---|---|---|---|---|---|---|---|
| 6-DOF GraspNet [7] | 10 / 10 | 4 / 10 | 10 / 10 | 5 / 10 | 10 / 10 | 2 / 10 | 10 / 10 | 3 / 10 | 54 / 80 (67.5%) |
| PoiNt-SE(3)-Dif [8] | 10 / 10 | 9 / 10 | 8 / 10 | 9 / 10 | 10 / 10 | 10 / 10 | 10 / 10 | 10 / 10 | 76 / 80 (95.0%) |
| EquiGraspFlow (Ours) | 10 / 10 | 9 / 10 | 10 / 10 | 9 / 10 | 10 / 10 | 10 / 10 | 10 / 10 | 10 / 10 | **78 / 80 (97.5%)** |

Table 2: Grasp success rates for various objects in real-world experiments. (R) indicates that objects are in rotated poses.

After capturing RGB-D data of the object from multiple viewpoints, the object's partial point clouds are fused to create a full point cloud, which is subsequently downsampled to 1,024 points. This data is passed through each model to generate 100 grasp poses. Following this, grasp poses that are infeasible due to the constraints such as the manipulator's workspace, singularities, and collisions with the environments are excluded. From the remaining feasible grasp poses, ten grasp poses are randomly selected to measure the success rate.

Table 2 shows the results of real-world grasping. Our model achieves performance comparable to that in simulation, demonstrating its seamless application to real-world tasks without sim-to-real issues. Even when using noisy real-world data, our model successfully generates grasp poses. Additionally, our model outperforms the baselines, similar to the simulation results.

## 6 Conclusion

In this paper, we introduce EquiGraspFlow, an $SE(3)$-equivariant 6-DoF grasp pose generative model. Our approach revolves around two key ideas: (i) constructing a framework for learning invariant conditional distributions on the $SE(3)$ manifold which is essential for equivariant grasp pose generation, and (ii) designing a novel equivariant lifting layer for our method. Unlike existing grasp pose generative models, our model ensures $SE(3)$-equivariance in generating grasp poses, resulting in improved performance and consistency. From simulation experiments, we conduct quantitative evaluations against baselines, demonstrating the superior grasp pose generation performance of our model. Furthermore, the consistent performance demonstrated across varying object rotations verifies the equivariance of the generated grasp poses and underscores the robustness of our method. Additionally, real-world experiments confirm the seamless applicability of our method to real-world scenarios, underscoring its practical relevance and effectiveness in real-world applications.

**Limitations and Future Works** Our model, trained on full point clouds, may struggle to generate accurate grasp poses when robots encounter occlusions that prevent full observation of an object. To address this issue, we conducted additional experiments on generating grasp poses from partial observations of objects, as detailed in Appendix C.2. The results of this experiment demonstrate that, even with partial point clouds, our method effectively generates grasp poses and outperforms the baselines. In future work, this model could be applied in real-world experiments to generate grasp poses in constrained environments where objects are difficult to observe from multiple viewpoints.

In the real-world experiments, we randomly selected ten grasp poses that provided statistically similar results to the generated grasp poses. However, if the objective is to identify the most promising grasp poses among the generated ones, one could leverage the probabilities of generated samples calculated from the CNFs or diffusion models. Selecting grasp poses with the highest probabilities could improve the success rate. This approach, however, has the drawback of primarily selecting grasp poses from regions with the highest probability, leading to a reduction in diversity.

Additionally, our model, which generates grasp pose without considering obstacles, requires a collision-checking algorithm for real-world grasping. However, by utilizing a dataset that includes grasp poses for a target object in cluttered environments, we can develop a collision-free grasp pose generative model without the need for additional modules. Incorporating equivariance into such a model is a direction for future work.

**Acknowledgments**

This work was supported in part by IITP-MSIT grant RS-2021-II212068 (SNU AI Innovation Hub), IITP-MSIT grant 2022-220480, RS-2022-II220480 (Training and Inference Methods for Goal Oriented AI Agents), MSIT(Ministry of Science, ICT), Korea, under the Global Research Support Program in the Digital Field program(RS-2024-00436680) supervised by the IITP(Institute for Information & Communications Technology Planning & Evaluation), KIAT grant P0020536 (HRD Program for Industrial Innovation), SRRC NRF grant RS-2023-00208052, SNU-AIIS, SNU-IPAI, SNU-IAMD, SNU BK21+ Program in Mechanical Engineering, SNU Institute for Engineering Research, and Microsoft Research Asia.

Yonghyeon Lee was the beneficiary of an individual grant from CAINS supported by a KIAS Individual Grant (AP092701) via the Center for AI and Natural Sciences at Korea Institute for Advanced Study.

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

# Appendix

## A  Proof of Proposition 1

In this section, we restate and prove Proposition 1 in Section 4.1.

**Proposition 1.** *Suppose a prior conditional distribution $p_0(T|\mathcal{P})$ is SE(3)-invariant. If the time-dependent angular and linear velocity fields $\omega, v$ are SE(3)-equivariant, then the transformed conditional distribution $p_t(T|\mathcal{P})$ at any time $t \geq 0$, defined via the flow of ODEs $\dot{R} = [\omega(t, \mathcal{P}, T)]R$ and $\dot{x} = v(t, \mathcal{P}, T)$, is SE(3)-invariant.*

To this end, we first introduce the concept of a conditional flow derived from the time-dependent conditional velocity fields and define SE(3)-equivariance of the conditional flow. Subsequently, we demonstrate that SE(3)-equivariant time-dependent conditional velocity fields induce an SE(3)-equivariant conditional flow. Finally, we prove Proposition 1 by establishing that, starting from an SE(3)-invariant prior conditional distribution, an SE(3)-equivariant conditional flow preserves the invariance over time.

### A.1  SE(3)-Equivariant Conditional Flows

Consider a trajectory on the SE(3) manifold, starting from an initial pose $T \in$ SE(3) and guided by the time-dependent angular and linear velocity fields, $\omega$ and $v$, conditioned on a point cloud $\mathcal{P}$. This trajectory is called an *integral curve* for $\omega$ and $v$ conditioned on $\mathcal{P}$ and starting at $T$, and is denoted by $\gamma : \mathbb{R} \to$ SE(3). By decomposing the SO(3) and $\mathbb{R}^3$ components such that $\gamma(t) = (\gamma_R(t), \gamma_x(t))$, the integral curve is defined via the following ordinary differential equations (ODEs):

$$\dot{\gamma}_R(t) = [\omega(t, \mathcal{P}, \gamma(t))]\gamma_R(t), \quad \dot{\gamma}_x(t) = v(t, \mathcal{P}, \gamma(t)), \quad \gamma(0) = T. \tag{3}$$

A *conditional flow* of the velocity fields $\omega$ and $v$ conditioned on $\mathcal{P}$ is defined as a mapping $f(t, \mathcal{P}, T) \in$ SE(3), where $t \in \mathbb{R}$ and $T \in$ SE(3) represent time and the initial pose, respectively. Here, $f(t, \mathcal{P}, T) = \gamma(t)$ where $\gamma$ is the integral curve for $\omega$ and $v$ conditioned on $\mathcal{P}$ and starting at $T$.

Now, we define the SE(3)-equivariance of a conditional flow as follows:

**Definition 3.** *A flow on SE(3) conditioned on a point cloud, denoted by $f(t, \mathcal{P}, T)$, is SE(3)-equivariant if $f(t, T'\mathcal{P}, T'T) = T'f(t, \mathcal{P}, T)$ for any $T' \in$ SE(3).*

Next, We demonstrate that SE(3)-equivariant time-dependent conditional velocity fields induce the SE(3)-equivariance of their conditional flow through the following Proposition:

**Proposition 2.** *For any time-dependent conditional angular and linear velocity fields $\omega$ and $v$, their conditional flow $f$ is SE(3)-equivariant if $\omega$ and $v$ are SE(3)-equivariant.*

*Proof.* Consider an arbitrary point cloud $\mathcal{P}$ and fix initial pose $T$ as an arbitrary element in SE(3). Then, $f(t, \mathcal{P}, T) = \gamma(t) = (\gamma_R(t), \gamma_x(t))$ represents the integral curve for $\omega, v$ conditioned on $\mathcal{P}$ and starting at $T$. The ODEs governing this integral curve are given by the same equations as (3).

For any $T' = (R', x') \in$ SE(3), $f(t, T'\mathcal{P}, T'T) = \tilde{\gamma}(t) = (\tilde{\gamma}_R(t), \tilde{\gamma}_x(t))$ where $\tilde{\gamma}$ is the integral curve for $\omega, v$ conditioned on $T'\mathcal{P}$ and starting at $T'T$. The ODEs for this integral curve are given by:

$$\dot{\tilde{\gamma}}_R(t) = [\omega(t, T'\mathcal{P}, \tilde{\gamma}(t))]\tilde{\gamma}_R(t), \quad \dot{\tilde{\gamma}}_x(t) = v(t, T'\mathcal{P}, \tilde{\gamma}(t)), \quad \tilde{\gamma}(0) = T'T. \tag{4}$$

Now, consider an integral curve $\hat{\gamma}$ defined as $\hat{\gamma}(t) = (\hat{\gamma}_R(t), \hat{\gamma}_x(t)) := (R'\gamma_R(t), R'\gamma_x(t) + x') = T'(\gamma_R(t), \gamma_x(t)) = T'\gamma(t) = T'f(t, \mathcal{P}, T)$. This integral curve results from transforming the integral curve $\gamma(t)$ by $T'$.

To prove the SE(3)-equivariance of the conditional flow, we need to show that $\tilde{\gamma}$ and $\hat{\gamma}$ are the same integral curve. Specifically, we need to show that $\tilde{\gamma}(t) = f(t, T'\mathcal{P}, T'T) = T'f(t, \mathcal{P}, T) = \hat{\gamma}(t)$.

Noting that $R[a]R^T = [Ra]$ for any $R \in \mathrm{SO}(3)$ and $a \in \mathbb{R}^3$, we analyze $\dot{\hat{\gamma}}_R(t)$ as follows:

$$
\begin{aligned}
\dot{\hat{\gamma}}_R(t) &= \frac{d}{dt}(R'\gamma_R(t)) \\
&= R'\dot{\gamma}_R(t) \\
&= R'[\omega(t, \mathcal{P}, \gamma(t))]\gamma_R(t) \\
&= [R'\omega(t, \mathcal{P}, \gamma(t))]R'\gamma_R(t) \\
&= [\omega(t, T'\mathcal{P}, T'\gamma(t))]R'\gamma_R(t) \\
&= [\omega(t, T'\mathcal{P}, \hat{\gamma}(t))]\hat{\gamma}_R(t).
\end{aligned}
\tag{5}
$$

Similarly, for $\hat{\gamma}_x(t)$, we have:

$$
\begin{aligned}
\dot{\hat{\gamma}}_x(t) &= \frac{d}{dt}(R'\gamma_x(t) + x') \\
&= R'\dot{\gamma}_x(t) \\
&= R'v(t, \mathcal{P}, \gamma(t)) \\
&= v(t, T'\mathcal{P}, T'\gamma(t)) \\
&= v(t, T'\mathcal{P}, \hat{\gamma}(t)).
\end{aligned}
\tag{6}
$$

Finally, note that $\hat{\gamma}(0) = T'\gamma(0) = T'T$. Thus, $\tilde{\gamma}(t)$ and $\hat{\gamma}(t)$ satisfy the same ODEs, and the uniqueness of the solution of the ODE ensures that $\tilde{\gamma}$ and $\hat{\gamma}$ are the same integral curve. Consequently, we have $f(t, T'\mathcal{P}, T'T) = T'f(t, \mathcal{P}, T)$ for any $T' \in \mathrm{SE}(3)$, demonstrating that $f$ is $\mathrm{SE}(3)$-equivariant. $\qquad\square$

## A.2   SE(3)-Invariant Conditional Distributions

To demonstrate that an $\mathrm{SE}(3)$-equivariant conditional flow preserves the invariance of an $\mathrm{SE}(3)$-invariant prior conditional distribution, we present the following proposition.

**Proposition 3.** *Suppose a prior conditional distribution $p_0(T|\mathcal{P})$ is $\mathrm{SE}(3)$-invariant. If a conditional flow $f$ is $\mathrm{SE}(3)$-equivariant, then the transformed conditional distribution $p_t(T|\mathcal{P})$ at any time $t \geq 0$ defined via the flow is $\mathrm{SE}(3)$-invariant.*

*Proof.* To prove this proposition, we first extend Theorem 3 from [25], which involves a general Riemannian manifold and a general group, to a conditional version.

Consider a Riemannian manifold $(\mathcal{M}, h)$ with a group $G$. Denote the action of an element $g \in G$ on $\mathcal{M}$ by the map $L_g : \mathcal{M} \to \mathcal{M}$. The map $L_g$ is isometric if, for any tangent vectors $u$ and $v$ at any point $x \in \mathcal{M}$, the following condition holds: $h(d(L_g)_x(u), d(L_g)_x(v)) = h(u, v)$, where $d(L_g)_x$ represents the differential of $L_g$ at $x$. If $L_g$ is isometric, then $\left|\det J_{L_g}(x)\right| = 1$ for any $x \in \mathcal{M}$, where $J_{L_g}(x)$ denotes the Jacobian matrix of the map $L_g$ evaluated at $x$ and expressed in local coordinates.

Let $c$ denote a condition variable. The conditional flow at time $t$ is represented by the map $f_{t,c} : \mathcal{M} \to \mathcal{M}$. This flow transforms a prior conditional distribution $p_0(x|c)$ into the conditional distribution $p_t(x|c)$. The likelihood of the transformed conditional distribution is given by the following change of variables formula:

$$
p_t(x|c) = p_0\left(f_{t,c}^{-1}(x)\big|c\right)\left|\det J_{f_{t,c}^{-1}}(x)\right|.
\tag{7}
$$

Assuming the action of $g \in G$ on $c$ is well-defined and denoted by $g \cdot c$, a conditional distribution $p(x|c)$ is $G$-invariant if $p(L_g(x)|g \cdot c) = p(x|c)$ for any $g \in G$. The conditional flow $f_{t,c}$ is $G$-equivariant if, $f_{t,g \cdot c}(L_g(x)) = L_g(f_{t,c}(x))$ for any $g \in G$, i.e., $f_{t,g \cdot c} \circ L_g = L_g \cdot f_{t,c}$ and $L_g^{-1} \circ f_{t,g \cdot c}^{-1} = f_{t,c}^{-1} \circ L_g^{-1}$.

Assuming that the map $L_g$ is isometric for any $g \in G$, we can prove Proposition 3 in a general Riemannian manifold $\mathcal{M}$ and an isometric group $G$ as follows:

$$p_t(L_g(x)|g \cdot c)$$

$$= p_0\left(f_{t,g\cdot c}^{-1}(L_g(x))|g \cdot c\right)\left|\det J_{f_{t,g\cdot c}^{-1}}(L_g(x))\right|$$

$$= p_0\left(L_{g^{-1}}\left(f_{t,g\cdot c}^{-1}(L_g(x))\right)|c\right)\left|\det J_{f_{t,g\cdot c}^{-1}}(L_g(x))\right| \qquad \text{(invariant prior)}$$

$$= p_0\left(\left(L_{g^{-1}} \circ f_{t,g\cdot c}^{-1} \circ L_g\right)(x)|c\right)$$
$$\underbrace{\left|\det J_{L_{g^{-1}}}\left(\left(f_{t,g\cdot c}^{-1} \circ L_g\right)(x)\right)\right|}_{=1}\left|\det J_{f_{t,g\cdot c}^{-1}}(L_g(x))\right|\underbrace{\left|\det J_{L_g}(x)\right|}_{=1}$$

$$= p_0\left(\left(L_{g^{-1}} \circ f_{t,g\cdot c}^{-1} \circ L_g\right)(x)|c\right) \qquad\qquad\qquad (8)$$
$$\left|\det J_{L_{g^{-1}}}\left(\left(f_{t,g\cdot c}^{-1} \circ L_g\right)(x)\right)J_{f_{t,g\cdot c}^{-1}}(L_g(x))J_{L_g}(x)\right| \qquad \text{(multiplicativity)}$$

$$= p_0\left(\left(L_{g^{-1}} \circ f_{t,g\cdot c}^{-1} \circ L_g\right)(x)|c\right)\left|\det J_{L_{g^{-1}} \circ f_{t,g\cdot c}^{-1} \circ L_g}(x)\right| \qquad \text{(chain rule)}$$

$$= p_0\left(\left(L_g^{-1} \circ f_{t,g\cdot c}^{-1} \circ L_g\right)(x)|c\right)\left|\det J_{L_g^{-1} \circ f_{t,g\cdot c}^{-1} \circ L_g}(x)\right| \qquad (L_{g^{-1}} = L_g^{-1})$$

$$= p_0\left(\left(f_{t,c}^{-1} \circ L_g^{-1} \circ L_g\right)(x)|c\right)\left|\det J_{f_{t,c}^{-1} \circ L_g^{-1} \circ L_g}(x)\right| \qquad \text{(equivariant flow)}$$

$$= p_0\left(f_{t,c}^{-1}(x)|c\right)\left|\det J_{f_{t,c}^{-1}}(x)\right|$$

$$= p_t(x|c).$$

Proposition 3 is a special case where $\mathcal{M} = \mathrm{SE}(3)$, $G = \mathrm{SE}(3)$, $c = \mathcal{P}$, and the group action of $T' \in \mathrm{SE}(3)$ on $T \in \mathrm{SE}(3)$, denote by $L_{T'}(T) = T'T$, is the left translation map which is isometric. Hence, Proposition 3 is proved. $\qquad\square$

We now prove Proposition 1 by utilizing Proposition 2 and Proposition 3.

*Proof of Proposition 1.* Since the angular and linear velocity fields $\omega$ and $v$ are SE(3)-equivariant, it follows from Proposition 2 that their flow $f$ is SE(3)-equivariant. Thus, by Proposition 3, the conditional distribution $p_t(T|\mathcal{P})$, defined via the flow of the velocity fields, is SE(3)-invariant. $\qquad\square$

## B  Implementation Details

### B.1  Details for Networks

To model the time-dependent conditional velocity fields $\omega_\theta(t, \mathcal{P}, T)$ and $v_\phi(t, \mathcal{P}, T)$ with SO(3)-equivariance, we employ Vector Neuron (VN) architectures [16], which are specifically designed for SO(3)-equivariance. The structure of these velocity fields is illustrated in Figure 8.

To encode the point cloud $\mathcal{P}$, we utilize the backbone of the VN-DGCNN designed for classification tasks, up to the invariant layer, excluding batch normalization layers, and adding a 170-size EdgeConv module at the sixth module position. A mean pooling layer is then applied to pool the point dimension, extracting a representation $z$ of 341 three-dimensional vectors. The grasp pose $T$ is reconfigured by concatenating the three rotation column vectors and the translation vector, resulting in four three-dimensional vectors. Time $t$ is converted into a three-dimensional vector through the lifting layer. The VN-MLP then concatenates these list of three-dimensional vectors as input and outputs the angular and linear velocities. The VN-MLP consists of five hidden VN-Linear layers, each followed by VN-LeakyReLU activation with a negative slope 0.2, and one output VN-Linear layer. The sizes of the hidden layers are (256, 256, 128, 128, 128), and the output layer size is 2, corresponding to two three-dimensional velocity vectors.

The lifting layer uses the representation $z$ and the grasp pose $T$ to convert the scalar time $t$ into a three-dimensional vector. This process involves a VN-Linear layer with a size of 1, producing an

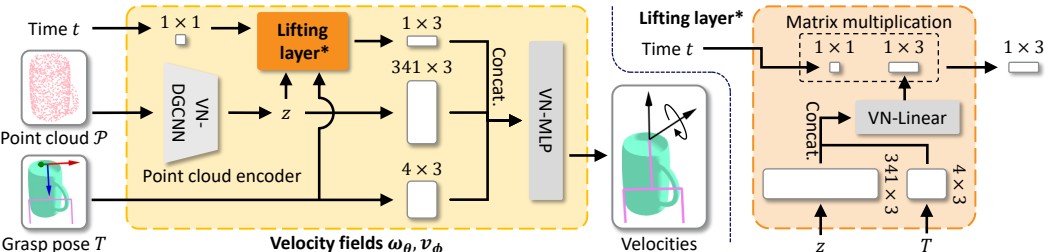

Figure 8: Structure of the velocity fields and the lifting layer. The VN-DGCNN encodes the point cloud $\mathcal{P}$ into a representation $z$ consisting of 341 three-dimensional vectors. The VN-Linear layer in the lifting layer uses this representation $z$ along with the grasp pose $T$, to produce a matrix of size $1 \times 3$. This matrix lifts the time variable $t$ (of size $1 \times 1$) to a three-dimensional vector. Finally, the VN-MLP in the velocity fields takes as input the concatenated list of the lifted time, representation, and grasp pose, and outputs the angular and linear velocities.

output matrix of size $1 \times 3$. This output matrix is then multiplied to the time variable $t$ (size $1 \times 1$), resulting in a single three-dimensional vector.

## B.2 Details for Training and Sampling

**Dataset** We use a dataset comprising 175 laptops, 94 mugs, 38 bowls, and 82 pencils obtained from the ACRONYM dataset [6]. The number of objects per each category and each dataset split are detailed in Table 3. For each category – laptop, mug, bowl, and pencil – an average of 1,643, 1,124, 1,747, and 1,737 6-DoF grasp poses are provided, respectively.

|  | Laptop | Mug | Bowl | Pencil |
|---|---|---|---|---|
| training | 105 | 54 | 21 | 50 |
| validation | 35 | 18 | 7 | 16 |
| test | 35 | 22 | 10 | 16 |

Table 3: Number of objects per each object category and each dataset split.

**Flow Matching** We employ the Flow Matching (FM) framework [29, 30] to train our CNF model. The core element of FM involves designing the *per-sample* target vector field $u_t^*(T|T_1)$ and the corresponding probability path $p_t(T|T_1)$, where $T_1 = (R_1, x_1)$ represents a particular sample from the target distribution $q(T|\mathcal{P})$. In our approach, we separate the rotation and translation components in $u_t^*(T|T_1) = (\omega_t^*(R|R_1), v_t^*(x|x_1))$. We then define the target angular and linear velocity fields $\omega_t^*(R|R_1)$ and $v_t^*(x|x_1)$ as follows:

$$[\omega_t^*(R|R_1)] = \frac{\log(R^T R_1)}{1-t}, \quad v_t^*(x|x_1) = \frac{x_1 - x}{1-t}. \tag{9}$$

Consequently, the training objective for EquiGraspFlow is designed as

$$\mathcal{L} = \mathbb{E}_{t, T_1 \sim q(T|\mathcal{P}), T \sim p_t(T|T_1)} \left[ \frac{1}{2} \left\| [\omega_\theta(t, \mathcal{P}, T)] - \frac{\log(R^T R_1)}{1-t} \right\|_F^2 + \left\| v_\phi(t, \mathcal{P}, T) - \frac{x_1 - x}{1-t} \right\|^2 \right] \tag{10}$$

where $|| \cdot ||_F$ denotes the Frobenius norm, and $T = (R, x)$ and $T_1 = (R_1, x_1)$.

One thing to note is that it might seem natural to design the vector field on the $SE(3)$ manifold instead of separating the rotation and translation components, similarly to how we design the angular velocity field on the $SO(3)$ manifold as shown in (9). However, this approach results in screw motion-shaped paths of grasp poses, where the translation may not follow a straight line toward the target grasp pose. In the context of our grasp pose generation task, separating the rotation and translation and ensuring that the translation motion directly heads toward the target grasp pose is a more intuitive and appropriate vector field formulation.

**Guided Flows** Guided Flows [31] is a technique that enhances the sample quality and efficiency of conditional generative models by integrating classifier-free guidance [38] into Flow Matching framework. This method employs a guided velocity fields during sampling, defined as a weighted sum of unconditional and conditional velocity fields. Using an empty set $\varnothing$ as a null condition for

the point cloud input, we define the guided angular and linear velocity fields $\tilde{\omega}_\theta$ and $\tilde{v}_\phi$ as follows, utilizing the weight parameter $\beta$:

$$\tilde{\omega}_\theta(t, \mathcal{P}, T) = (1 - \beta)\omega_\theta(t, \varnothing, T) + \beta\omega_\theta(t, \mathcal{P}, T),$$
$$\tilde{v}_\phi(t, \mathcal{P}, T) = (1 - \beta)v_\phi(t, \varnothing, T) + \beta v_\phi(t, \mathcal{P}, T). \tag{11}$$

When $\varnothing$ is input, the point cloud encoder outputs a list of zero vectors as $z$. The guided velocity fields remain $\mathrm{SE}(3)$-equivariant, as they are a linear combination of unconditional and conditional velocity fields, both of which are $\mathrm{SE}(3)$-equivariant. To train the unconditional velocity fields, we randomly replace $\mathcal{P}$ with the empty set $\varnothing$ with a probability of 20% during training. For sampling, we use $\beta = 2$ to evaluate average performance and $\beta = 2.5$ to assess the consistency of performance.

**Optimizer**   Adam optimizer [39] with learning rate $1 \times 10^{-4}$ is utilized to train the baselines and our model. L2 regularization with hyperparameter $1 \times 10^{-5}$ is employed for training EquiGraspFlow.

**Sampling**   The fourth-order Runge-Kutta MK method on Lie groups [40] is utilized as the ODE solver, with 20 steps employed.

### B.3   Details for Grasping Experiments

**Grasping in Simulation**   In the simulation experiments, the gripper and object float without any obstacles. After grasping the object, the gripper is shaken to robustly assess the success of the grasp.

**Grasping in Real-World**   For the real-world experiments, a Franka Panda gripper is equipped with a 7-DoF Franka Emika Panda robot, and the object is placed on a table. The object's point cloud is obtained using an Intel RealSense Depth Camera D435 mounted on the gripper. The depth camera captures an RGB-D image of the scene, and the object is segmented using Language Segment-Anything [41], which is built on the Segment Anything Model [42] and Grounding DINO [43], resulting in a partial point cloud of the object. By combining partial point clouds from multiple viewpoints, we reconstruct the full point cloud of the object. We then downsample the full point cloud to a uniformly downsampled point cloud with 1,024 points using voxel downsampling.

The robot motion for grasping an object in real-world experiments is designed as follows. To prevent collisions with the object during the movement of the gripper toward the generated grasp pose, we first move the gripper to a pre-grasp pose. This pre-grasp pose is offset from the grasp pose by a small distance in the $-z$ direction in the gripper's frame (the $z$-axis of the gripper's frame represents the direction of gripper's palm). Next, we move the gripper to the grasp pose and execute the grasp. Once the object is grasped, the gripper is lifted by small distance. The success of the grasp is manually determined based on whether the object is held securely by the gripper. After each grasping attempt, we manually reset the position and orientation of the object to its initial state.

## C   Additional Experiments and Results

### C.1   Sampling Time

We conduct an experiment to measure the sampling time of the baselines and EquiGraspFlow. Each model generates 100 grasp poses from a point cloud with 1024 points. Sampling is conducted on a single NVIDIA GeForce RTX 3090 GPU, and results are averaged over 100 experiments.

| Model | Sampling time (ms) |
|---|---|
| 6-DOF GraspNet [7] | 41.08 |
| PoiNt-SE(3)-Dif [8] | 1666.56 |
| EquiGraspFlow (Ours) | 256.76 |

Table 4: Sampling times for generating 100 grasp poses from a point cloud consisting of 1024 points.

Table 4 shows the average sampling time of each model. While 6-DOF GraspNet [7] is about six times faster than EquiGraspFlow, its grasping performance is less-than-desirable. On the other hand, EquiGraspFlow is not only about six times faster but also shows better grasping performance than PoiNt-SE(3)-Dif [8].

| Augmentation strategy | None | | | | SO(3)-aug | | | |
|---|---|---|---|---|---|---|---|---|
| Object category | Laptop | Mug | Bowl | Pencil | Laptop | Mug | Bowl | Pencil |
| 6-DOF GraspNet [7] | 0.8778 | 0.8726 | 1.0766 | 0.7241 | 0.6012 | 0.7628 | 0.8724 | 0.4233 |
| PoiNt-SE(3)-Dif [8] | 0.7875 | 0.8892 | 1.0407 | 0.5957 | 0.5851 | 0.7186 | 0.4733 | 0.4498 |
| EquiGraspFlow (Ours) | **0.3564** | **0.4801** | **0.3204** | **0.2879** | **0.3032** | **0.4086** | **0.2675** | **0.2648** |

(a) Average EMD (lower values are better)

| Augmentation strategy | None | | | | SO(3)-aug | | | |
|---|---|---|---|---|---|---|---|---|
| Object category | Laptop | Mug | Bowl | Pencil | Laptop | Mug | Bowl | Pencil |
| 6-DOF GraspNet [7] | 7.31 | 12.16 | 5.42 | 17.47 | 17.18 | 24.99 | 18.89 | 60.97 |
| PoiNt-SE(3)-Dif [8] | 14.50 | 20.69 | 12.84 | 34.86 | 87.75 | 50.12 | 95.82 | 97.44 |
| EquiGraspFlow (Ours) | **95.86** | **84.91** | **100.00** | **99.64** | **96.11** | **84.10** | **99.87** | **99.53** |

(b) Average grasp success rate (%) (higher values are better)

Table 5: EMD and grasp success rate of the grasp poses generated from partial point cloud.

## C.2   Grasp Pose Generation from Partial Point Cloud

To evaluate the performance of grasping in scenarios where objects cannot be fully observed from all viewpoints, we train and test the baselines and EquiGraspFlow when partial point clouds are input. Instead of uniformly sampling points from the mesh surface, we obtain a partial point cloud by sampling 512 points only from the mesh surface visible from a viewpoint defined by a 3D unit vector originating from the object's center. All grasp poses in the dataset are utilized regardless of the viewpoint, requiring the model to generate grasp poses even for unobserved parts of the object.

The data augmentation for object rotation follows the same approach as in the full point cloud experiments, utilizing either the *None* or SO(3)-*aug* strategy in the training dataset, while the validation and test datasets utilize evenly sampled rotations using Super-Fibonacci Spirals algorithm [35]. The viewpoint vector on $S^2$ is randomly sampled in the training dataset, and is evenly sampled using Fibonacci lattice algorithm [44] in the validation and test datasets. Each object in the validation and test datasets is augmented with three rotations, and each rotated object is further augmented with three viewpoints. The grasp success rate is measured in Nvidia Isaac Gym simulator [37] using 25 generated grasp poses.

The average Earth Mover's Distance (EMD) and grasp success rate under different augmentation strategies are presented in Table 5. Despite a slight performance degradation from observing only portions of objects, our method still demonstrates strong performance, indicating its applicability in scenarios where objects are partially observed. Additionally, similar to the full point cloud experiments, EquiGraspFlow achieves lower EMD and higher grasp success rate compared to the baselines in both the *None* and SO(3)-*aug* settings. This demonstrates superior performance in learning grasp pose distribution and generating graspable poses even when objects are partially observed. Especially, EquiGraspFlow demonstrates a high grasp success rate even for mugs, which have complex shapes and are challenging for generating graspable poses under partial observation. This is attributed to the enhanced data efficiency resulting from the incorporation of SO(3)-equivariance, which enables the model to better learn how to grasp mugs from partial observation compared to the baselines.

## C.3   Grasp Pose Generation from Full Point Cloud

### C.3.1   Consistency with Respect to the Number of Rotations

We conduct additional experiments to investigate how performance consistency varies with the number of rotations. Using the same setup in the consistency experiments described in Section 5.2, we augment the test dataset with 2, 4, 6, and 8 rotations, respectively.

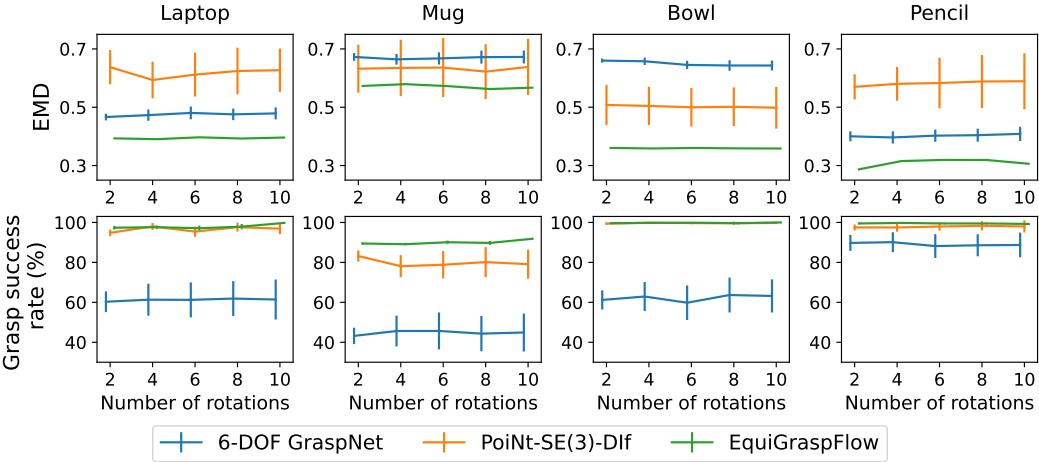

Figure 9: Average and standard deviation of EMD (lower values are better) and grasp success rate (higher values are better) with respect to the number of rotations. The standard deviation is calculated with respect to the object's rotation and then averaged across all objects.

Figure 9 presents an error bar plot illustrating the changes in the average and standard deviation of EMD and grasp success rate across different numbers of rotations for each object category. As indicated by the increasing size of the error bars, the baselines exhibit less consistency in grasp pose distribution and grasping performance as the number of rotations increases. This suggests that the more diverse the rotations, the less consistent the baselines are in generating grasp poses. In contrast, EquiGraspFlow exhibits zero standard deviations in EMD values across all number of rotations, indicating that our model generates a consistent grasp pose distribution. This consistency results from the $\mathrm{SO}(3)$-equivariance, which ensures the generation of equivariant grasp poses relative to the object's rotation. Although there are slight variations in the grasp success rate for EquiGraspFlow in some cases, these are attributed to the simulator's non-equivariance. As these deviations remain close to zero across all rotation counts, they further underscore the robustness of our model.

### C.3.2 Additional Visualizations

Figures 10 to 13 present additional visualizations of the generated grasp poses. These figures show the generated grasp poses of a laptop, mug, bowl and pencil across ten object rotations, along with the EMD and grasp success rate values. The object's point cloud is rotated and input into each model; however, in these figures, both the objects and the generated grasp poses are inversely rotated to align all scenes. Successful and failed grasp poses are indicated in green and red, respectively.

For all object types, the grasp poses generated by EquiGraspFlow are widely distributed across various parts of the objects, demonstrating that our model generates more diverse grasp poses compared to the baselines. For example, while PoiNt-SE(3)-Dif tends to concentrate generated grasp poses on specific parts of the laptop and bowl for certain object rotations, EquiGraspFlow distributes the grasp poses evenly across the objects regardless of their rotation. Additionally, PoiNt-SE(3)-Dif struggles to generate grasp poses that target the handle of the mug, whereas EquiGraspFlow generates grasp poses that are evenly distributed between the handle and the body of the mug.

Furthermore, the variance in the EMD and grasp success rate values indicates that EquiGraspFlow exhibits more consistent results across different object rotations. While the baselines show variability in grasp pose distribution and grasping performance depending on the object's rotation, our model maintains identical values across the ten object rotations, demonstrating the perfect equivariance of our approach.

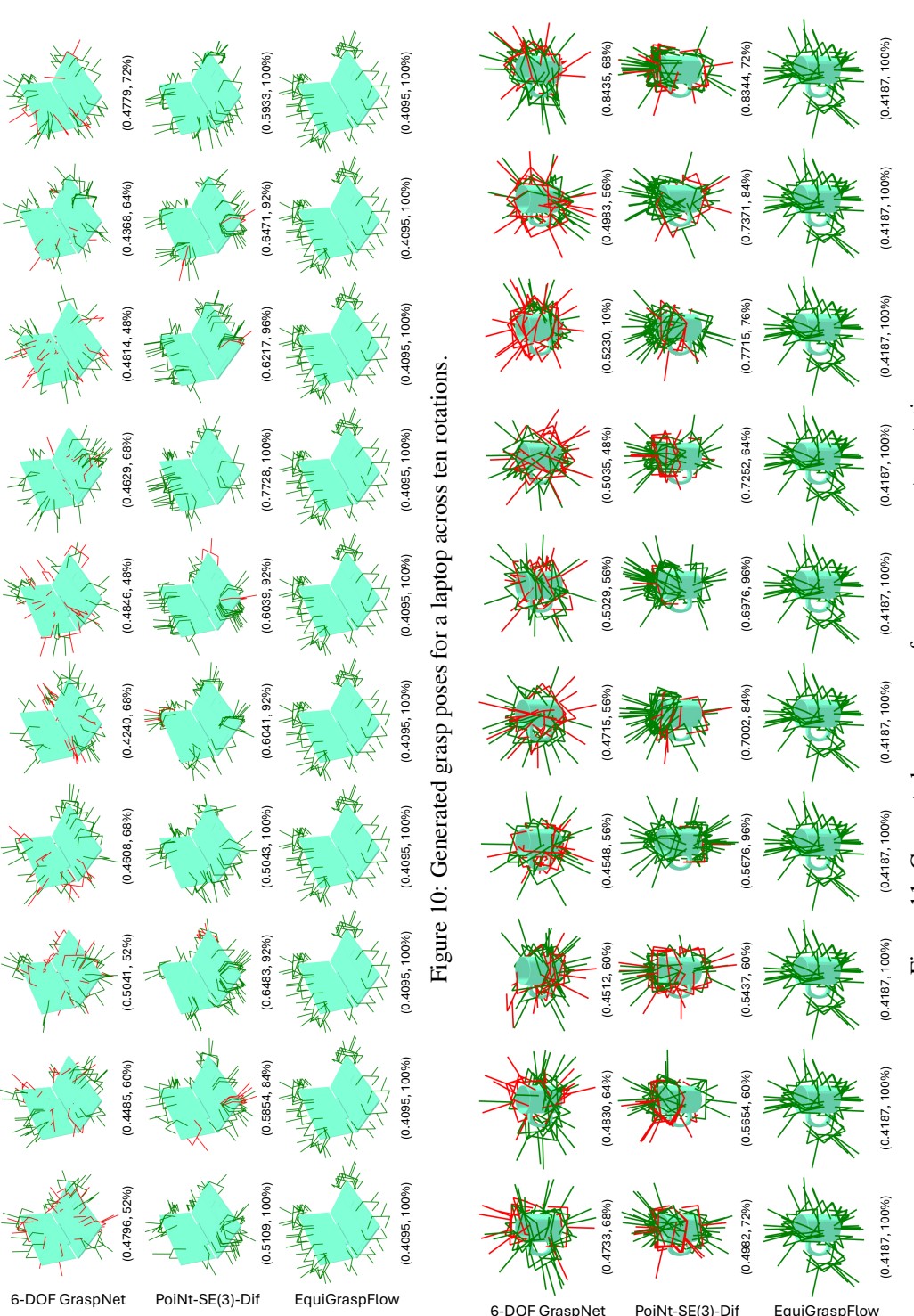

Figure 10: Generated grasp poses for a laptop across ten rotations.

Figure 11: Generated grasp poses for a mug across ten rotations.

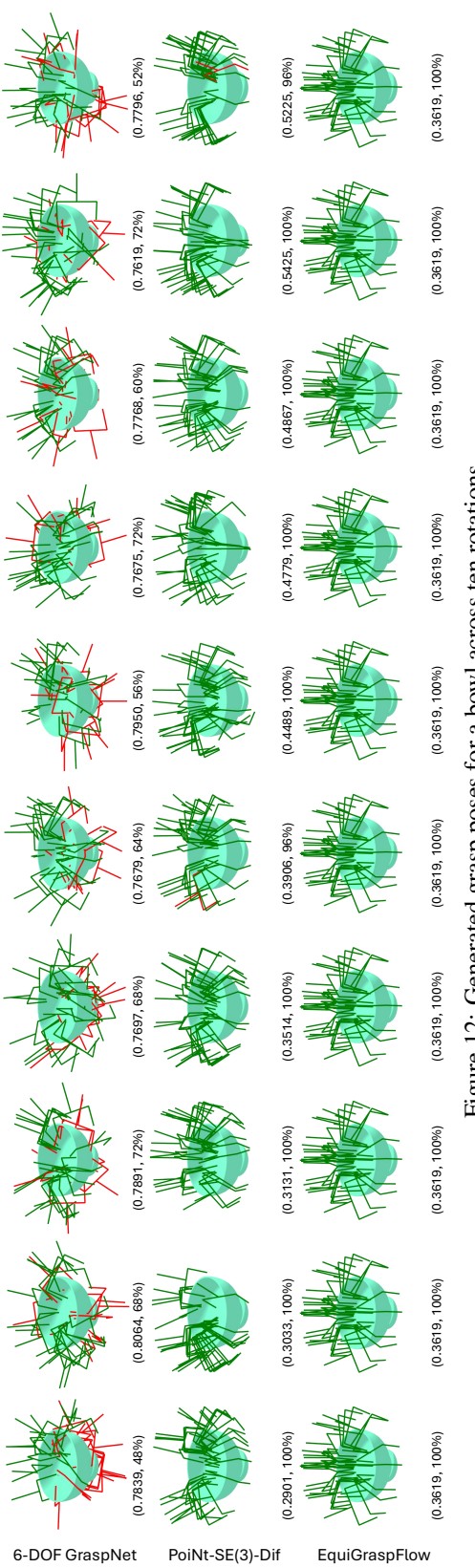

Figure 12: Generated grasp poses for a bowl across ten rotations.

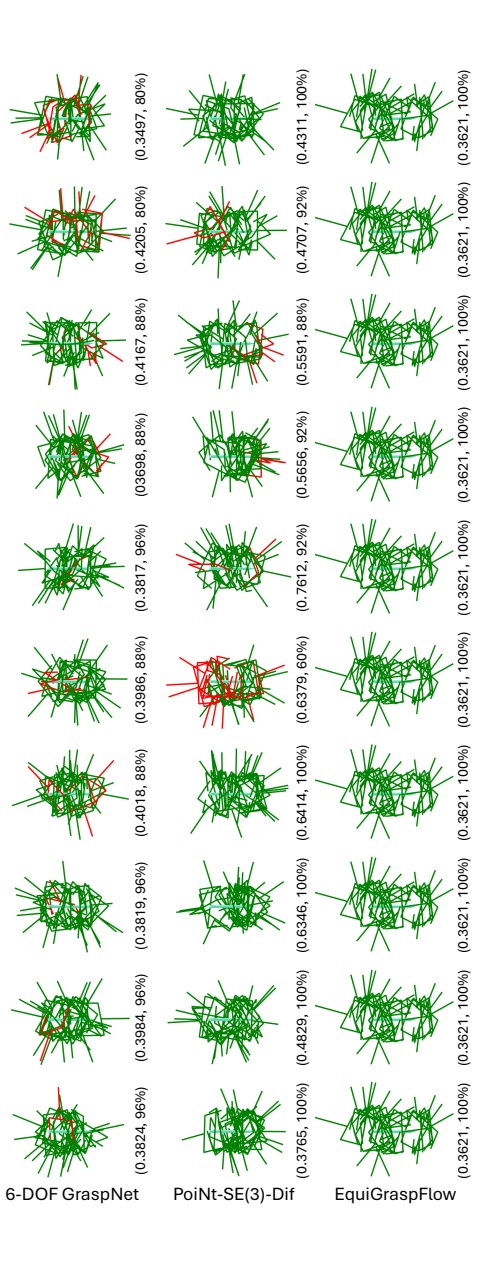

Figure 13: Generated grasp poses for a pencil across ten rotations.

