# OpenReview forum: "EquiGraspFlow: SE(3)-Equivariant 6-DoF Grasp Pose Generative Flows"
_robot-learning.org/CoRL/2024/Conference — CoRL 2024_

### Official Review · Reviewer_2vdQ · 2024-06-26

**Originality:** 3
**Technical Quality:** 4
**Clarity Of Presentation:** 4
**Potential Impact:** 3
**Recommendation:** 4
**Confidence:** 4

**Review:**

# Strengths
The idea of incorporating invariance in the generated distribution is original and sound in the context of grasp pose generation.

The paper is extremely well-written and well-organized. In particular, the relevant mathematics is introduced succinctly, with an elegant and clear formalism.

The experimental section is thorough. It contains both simulated and real-world experiments, evaluated via probabilistic distances (geodesic earth-mover distance) and downstream tasks (grasp success rate).


# Weaknesses
The main theoretical result (Proposition 1) is a basic and well-known mathematical fact, even in the applied machine learning literature. It holds, with the same simple proof, on any (compact) Lie group: the flow of an equivariant vector field on a Lie group $G$ transports $G$-invariant distributions to $G$-invariant ones. Similarly to this work, the result is used in several papers on group-invariant flow-based generative modelling – see, for example, Theorem 1 and Theorem 2 in [1], or Theorem 1 in [2], or Proposition 1 in [3]. I believe the authors should stress out that the result is not novel, and is a standard argument in the field (with appropriate references).

[1] Köhler et al., Equivariant Flows: Exact Likelihood Generative Learning for Symmetric Densities, ICML 2020.

[2] Zwartsenberg et al., Conditional Permutation Invariant Flows, TMLR 2023.

[3] Lu et al., Structure Preserving Diffusion Models, 2024.

# Additional Comments
-Line 18: The acronym DoF (degrees of freedom) is never introduced explicitly – I suggest doing it here. Even though it is standard terminology, it is customary to introduce all the acronyms in a paper for better accessibility.

**Quality Of The Limitations Section:**

3

**Questions For Rebuttal:**

I have no questions for the rebuttal.

**Robotics Focus:**

4

**Summary Of Paper:**

The work proposes a flow-based $SE(3)$-equivariant generative model on $SE(3)$ that is conditioned on a point-cloud in $\mathbb{R}^3$. The method is applied to grasp pose generation.

**Summary Of Recommendation:**

I think that the work combines equivariant deep learning with flow-based generative modelling in a clever way for the purpose of grasp pose sampling. Besides the weakness above – which has a theoretical nature – the work deserves acceptance, in my opinion, for its methodological and applied contributions.

---

### Official Review · Reviewer_Qvx1 · 2024-07-20
**A Promising Approach for SE(3)-Equivariant 6-DoF Grasp Synthesis**

**Originality:** 4
**Technical Quality:** 4
**Clarity Of Presentation:** 3
**Potential Impact:** 4
**Recommendation:** 4
**Confidence:** 4

**Review:**

### Strengths
- The problem tackled by this work is of highly practical relevancy for the learning-based grasping community;
- The presentation of the paper is generally clear and easy to follow except for some minor, less-friendly words for readers without sufficient background knowledge: L54: "no augmentation" and L56: "time-dependent velocity fields."
- The authors propose a technically sound formulation of the necessary conditions to guarantee the SE(3)-equivariance on the conditional continuous normalizing flows;
- A practical solution for constructing SE(3)-equivariant time-dependent velocity fields based on Vector Neurons is introduced;
- Comprehensive baselines, including other generative models such as VAE, GAN, and Diffusion models, are included in the experimental comparison;

### Weaknesses
- Lack of discussion of discrete normalizing flows: why do the authors focus on the continuous version?
- For the proposed lifting layer, instead of constructing the proposed layer, how about simply replicating the time variable to construct the vector? The concern of the current design is that, after the matrix multiplication, will the time information be lost or distorted, which might harm the actual effect of the velocity field?
- Lack of a run-time analysis in the discussion or experiment part;
- Lack of evidence on more object categories with various geometries;
- Lack of an ablation study on how the performance degrades with increasing rotations across the baselines. This ablation study is informative in the sense of knowing the limits of different approaches.
- In Table 2, it would be more understandable if "percentage" could be used.
- Performance gain on the real-robot experiment seems quite limited, i.e., only 2 successful grasps more. An explanation or failure analysis for this would be nice;

**Quality Of The Limitations Section:**

3

**Questions For Rebuttal:**

See the list of weaknesses above.

**Robotics Focus:**

4

**Summary Of Paper:**

The authors devise a SE(3)-Equivariant continuous normalizing flow model for generative 6-DoF Grasp Synthesis. Such equivariance can help the model generalize well with less data-preprocessing such as data augmentation. To this end, the authors  introduced necessary conditions for constructing such flows and a lifting layer in the conditional velocity field networks. The superior performance is validated in both simulated and real-world experiments for two object categories.

**Summary Of Recommendation:**

For the technically sound appraoch and convincing results, I vote for weak accept while looking forward to the manuscript with the points above addressed.

---

### Official Review · Reviewer_jkBE · 2024-07-21
**Initial Review**

**Originality:** 4
**Technical Quality:** 4
**Clarity Of Presentation:** 4
**Potential Impact:** 4
**Recommendation:** 4
**Confidence:** 4

**Review:**

Overall, the paper is clear and well-written. Moreover, it nicely motivates why SE(3) equivariant grasp pose generation is desired and presents convincing methodological contributions regarding the grasp generation architecture and exploiting a flow-based generative model that fits the needs of the desiredata. Additionally, real robot experiments are provided.

I still have a few remarks I will list in the following.

1) In Section 3, the authors explain continuous normalizing flows in SE(3). In line 105, they define a time-dependent vector field on SE(3), which, however, only maps to a single three-dimensional vector. Shortly after it becomes evident that rotation and translation are separated. It would nevertheless be good if the authors made clear from the beginning that they need two three-dimensional vectors for the flow in full SE(3).

2) In Section 4, and in general, the authors have to start from some sort of prior distribution. However, I could not find which prior distribution the authors used in the paper. Since this is important information, it should be added.

3) In the last paragraph of Section 4.2 the authors explain how to construct the equivariant lifting layer. The authors could provide more details at this point of the paper. If I understand it right, this lifting layer consists of an additional VN-MLP network outputting just a single point which is then scaled by time. Is this intuition correct? If yes, adding more context at this point of the paper would help clarity.

4) In the experiment section, Table 1 could be improved. For me, it would have helped to provide names for the first three rows of the table, i.e., something like Metrics, Object, and Data Augmentation. The latter, I guess, would help a lot in understanding the table correctly, as I initially thought that None corresponds to no augmentation, i.e., only using the nominal object poses during evaluation. However, if I understand correctly, it is related to the data augmentation strategies only.

5) At the end of Section 5.2. the authors mention in lines 281-283 that their method shows zero standard deviation. I found this statement a bit confusing at first, and I think the paper would benefit in clarity if the definition of this standard deviation (which is provided in Fig. 5) would be also added in the main text.

6) In the real-world experiments, the authors generate a bunch of grasp poses and subsequently, from the remaining feasible grasp poses, randomly select 10. While this is a valid procedure, SE(3)-Dif fields, since it outputs the energy values corresponding to each grasp pose, would allow for a more informed grasp selection strategy. Would this additional consideration further improve the performance of SE(3)-dif? In line with this question, especially in the limitations sections, the authors could discuss a bit in more detail how, in practice for their proposed method they would select the best grasp pose that is to be executed on the robot in a pick-and-place experiment, as testing ten generated grasp poses is not something one would do in practice.

7) As mentioned by the authors in the real-world experiments description (lines 222-231) and also in the limitations section, their method relies on a full / maximally complete pointcloud of the objects. I think it could be interesting to further improve the paper by providing an ablation w.r.t. how sensitive the current approach is w.r.t. the pointcloud being complete. I.e., does the performance drop to 0% success rate if only a single view pointcloud measurement is used? I think this information is not needed in the main paper but could be interesting information for the Appendix.

8) (Very Minor Comment). Section 3 is a background / preliminaries section. I would prefer that this is clear from its heading already, so the authors might consider to rename the section title to Background: Continuous ...

**Quality Of The Limitations Section:**

3

**Questions For Rebuttal:**

- What is the initial distribution from which you start sampling the grasp poses?
- In the real-world experiment, would SE(3)-Dif benefit and slightly reduce the number of failures when selecting the 10 grasp poses according to the energy?
- What procedure would the authors propose in a real-world scenario when they can only select a single grasp from the generated ones? I.e., what alternative strategies could be implemented with the presented approach apart from sampling at random?
- How sensitive is the method w.r.t. the pointcloud being complete?

**Robotics Focus:**

4

**Summary Of Paper:**

In this paper, the authors present a new grasp generative model. Compared to prior works, their model is SE(3) equivariant by definition, which results in improved performance. This is achieved on a methodological level through an equivariant encoder (VN) and a flow-based generative model.

**Summary Of Recommendation:**

The paper is very well written and has substantial contributions, which are nicely showcased in the experiments. Moreover, the experimental results are convincing and fully support the claims made in the paper.

---

### Author Rebuttal · Authors · 2024-08-13

We appreciate the time and effort the reviewers have dedicated to providing their valuable feedback on our paper.
We have carefully considered all the comments and have made the necessary revisions to improve the paper.

The following key changes have been incorporated into the revised manuscript:

* We have moved the section on the grasping environment to Appendix B.3.

* We have conducted simulation experiments anew. The Acronym dataset contains anomalous data, such as objects that include a spoon inside a mug or fruits inside a bowl. Since our target is to generate grasp poses for individual objects, we excluded such data. We have also expanded the object categories to include laptops and pencils, in addition to mugs and bowls. Thus, we have changed Table 1, Figure 5, and Figure 6 in Section 5.2, and Figure 3-6 in Appendix C.3.2.

* We excluded GAN-style 6-DOF GraspNet in experiments, since it is unofficial version.

* We have added content to the limitation and future works section, introducing a method for selecting generated grasp poses based on the probabilities calculated by the model.

* We have changed the overall topic of Appendix C to "Additional Experiments and Results."

* We have added experiments on sampling time in Appendix C.1.

* We have included experiments on grasp pose generation from partial point cloud in Appendix C.2. Additionally, we have revised the limitations and future works section to refer these experiments.

* We have added experiments about consistency with respect to the number of rotations in Appendix C.3.1.

Additionally, we highlight the changed parts in the main paper and the appendix in $\color{blue}{\text{blue}}$.

---

### Decision · Program_Chairs · 2024-09-04

**Decision:**

Accept

**Comment:**

PRE REBUTTAL:

High level summary of reviews:

Strengths:

- All reviewers state that the paper is clear, well-written, and well-organized.
- Multiple viewers suggest that the paper is technically sound and presents convincing methodological contributions.
- Multiple reviewers highlight the novelty of the approach (the idea of incorporating invariance in the generated distribution).
- The experimental section is thorough with real-world experiments. Reviewer jkBE highlights that experimental results are convincing and fully support the claims made in the paper.

Weaknesses:

- It lacks discussion of discrete normalizing flows and evidence on more object categories.
- The performance gain on the real-robot experiment is limited.
- Reviewer 2vdQ suggests that the main theoretical result (Proposition 1) is a basic and well-known mathematical fact. This should be clarified in updated versions of the manuscript.
- Reviewers Qvx1 and jkBE request the authors clarify a few details of the paper. Please see their extensive comments for details (e.g. run-time analysis, performance on more object categories, etc).

POST REBUTTAL:

During the rebuttal the authors clarified a number of points made by the reviewers and added additional experiments. This is clearly a strong submission and is stronger following the rebuttal.